# An in-situ flow-tube system for direct measurement of $N_2O_5$ heterogeneous uptake coefficients in polluted environments

Weihao WANG[1], Zhe WANG[1], Chuan YU[1,2], Men XIA[1], Xiang PENG[1], Yan ZHOU[3], Dingli YUE[3], Yubo OU[3], Tao WANG[1]

5  [1]Department of Civil and Environmental Engineering, The Hong Kong Polytechnic University, Hong Kong, China
[2]Environment Research Institute, Shandong University, Ji'nan, Shandong, China
[3]Guangdong Environmental Monitoring Center, State Environmental Protection Key Laboratory of Regional Air Quality Monitoring, Guangzhou, China

*Correspondence to*: Zhe Wang (z.wang@polyu.edu.hk); Tao Wang (cetwang@polyu.edu.hk)

10  **Abstract.** The heterogeneous reactivity of dinitrogen pentoxide ($N_2O_5$) on ambient aerosols plays a key role in atmospheric fate of $NO_x$ and formation of secondary pollutants. To better understand the reactive uptake of $N_2O_5$ on complex ambient aerosols, an in-situ experimental approach to direct measurement of $N_2O_5$ uptake coefficient ($\gamma N_2O_5$) was developed for application in environments with high, variable ambient precursors. The method utilizes an aerosol flow-tube reactor coupled with an iterative chemical box model to derive $\gamma N_2O_5$ from the depletion of synthetically generated $N_2O_5$ when mixed with 15  ambient aerosols. Laboratory tests and model simulations were performed to characterize the system and the factors affecting $\gamma N_2O_5$, including mean residence time, wall loss variability with RH, and $N_2O_5$ formation and titration with high levels of $NO/NO_x/O_3$. The overall uncertainty was estimated to be 37%-40% at $\gamma N_2O_5$ of 0.03 for RH varying from 20% to 70%. The results indicate that this flow tube coupled with the iterative model method could be buffered to NO concentrations below 8 ppbv and against air mass fluctuations switching between aerosol and non-aerosol modes. The system was then deployed in 20  the field to test its applicability under conditions of high ambient $NO_2/O_3$ and fresh NO emission. The results demonstrate that the iterative model improved the accuracy of $\gamma N_2O_5$ calculations under polluted environments, and thus support the further field deployment of the system to study the impacts of heterogeneous $N_2O_5$ reactivity on photochemistry and aerosol formation.

## 1 Introduction

Dinitrogen pentoxide ($N_2O_5$) is a nocturnal reactive intermediate in the atmospheric oxidation of nitrogen oxides ($NO_x$), which 25  plays an important role in atmospheric photochemistry and the production of secondary pollutants (e.g., Chang et al., 2011). $N_2O_5$ is formed from the reaction of nitrogen dioxide ($NO_2$) and nitrate radical ($NO_3$). Because $NO_3$ is photolytically unstable, it (and therefore $N_2O_5$) only accumulates under dark conditions (i.e., at night). The heterogeneous reactions of $N_2O_5$ on aerosols have been recognized as a major sink for $NO_x$, affecting the atmospheric lifetime of $NO_x$ and the formation of ozone and other secondary pollutants (e.g., Brown et al., 2007; Wang et al., 2016). The heterogeneous $N_2O_5$ loss rate on aerosols ($k_{aerosols}$)

depends on the uptake coefficient of $N_2O_5$ ($\gamma N_2O_5$) and the available aerosol surface area, and can be expressed using Eq. (1) when the gas phase diffusion effect is negligible (Fuchs and Sutugin, 1971).

$$k_{aerosols} = \tfrac{1}{4}\, c_{N2O5}\, Sa\, \gamma N_2O_5 \qquad\qquad\qquad (1)$$

where $c_{N2O5}$ (m/s) is the mean molecular speed of $N_2O_5$ and Sa ($m^2/m^3$) is the aerosol surface area concentration. $\gamma N_2O_5$ is the
reaction probability that a $N_2O_5$ molecule collides with the aerosol surface resulting in net removal via reactions on aerosols. Because $\gamma N_2O_5$ is a critical parameter to determine $N_2O_5$ uptake on aerosols, it is necessary to develop reliable methods to measure it.

$\gamma N_2O_5$ has typically been determined in laboratory using different types of flow tube and reactors to measure the decay rate of $N_2O_5$ in the presence of pure inorganic and organic aerosols or mixed aerosols under different conditions (e.g., Thornton et al.,
2003; Tang et al., 2017 and references cited therein). The $\gamma N_2O_5$ has been shown to be highly dependent on aerosol composition, temperature and relative humidity; different parameterizations of varying degrees of complexity have thus been proposed to relate $\gamma N_2O_5$ to aerosol composition (Anttila et al., 2006; Bertram and Thornton, 2009; Davis et al., 2008; Evans and Jacob, 2005; Riemer et al., 2009). In ambient conditions, several methods have been developed to derive $\gamma N_2O_5$ directly from atmospheric concentrations of $N_2O_5$. Brown et al. (2007) utilized steady-state approximation of $NO_3$ and $N_2O_5$ to derive $\gamma N_2O_5$
based on the correlation of inverse $N_2O_5$ steady-state lifetime with $NO_2$ concentration and aerosol surface area; Phillips et al. (2016) assumed a conserved air mass and used the production rates of $NO_3^-$ and $ClNO_2$ to derive $\gamma N_2O_5$; Wagner et al. (2013) applied an iterative chemical box model to derive the appropriate $\gamma N_2O_5$ to match the predicted $N_2O_5$ concentration to the measured values with the assumption of the reaction time starting at sunset and with no interception of other $NO_x$ emissions.

Bertram et al. (2009a) introduced an approach to directly measure $\gamma N_2O_5$ on ambient aerosols by utilizing an entrained aerosol
flow reactor coupled with a chemical ionization mass spectrometer (CIMS). By switching sampling between filtered and unfiltered ambient air, the reactivity of $N_2O_5$ was determined based on a comparison of the pseudo-first-order loss rate of $N_2O_5$ in ambient air with and without aerosols. The loss rate of $N_2O_5$ to aerosols ($k_{aerosols}$) could be derived from the concentration ratio at the exit of the flow reactor, with the assumption that the wall loss of $N_2O_5$ is constant in the successive two measurements and that all losses are first-order (Bertram et al., 2009a):

$$k_{aerosols} = -\frac{1}{\Delta t}\ln\left(\frac{[N_2O_5]_{\Delta t}^{w/aerosols}}{[N_2O_5]_{\Delta t}^{wo/aerosols}}\right), \qquad\qquad\qquad (2)$$

where $\Delta t$ is the mean residence time in the flow tube reactor, and $[N_2O_5]_{\Delta t}$ is the $N_2O_5$ concentration measured at the exit of the flow reactor in the two modes (i.e. the presence and absence of aerosols). This flow tube apparatus was deployed at two urban sites in Boulder and one coastal site in La Jolla to measure $\gamma N_2O_5$ on ambient aerosols (Bertram et al., 2009b; Riedel et al., 2012). They found that the fluctuation of relative humidity (RH) and $NO_3$ reactivity (mainly dominated by NO) could lead

to great uncertainty in measured $\gamma N_2O_5$, and therefore applied some screening criteria, including only data with a RH fluctuation of less than 2% and NO concentration lower than 750 pptv. This constraint resulted in about 20% of the data being used for further analysis. It was necessary to adopt these criteria because only first-order loss is considered in the flow tube reactor and other reactions involving ambient NO, $NO_2$, and $O_3$ are not. The latter treatment is suitable when ambient concentrations are low and the air mass is relatively stable, but may be problematic in polluted environments with high fresh $NO_x$ emissions, high $O_3$ concentrations, and rapidly changing air mass.

Several recent studies have revealed active $N_2O_5$ heterogeneous process on aerosols at polluted sites and its significant impacts on photochemistry and secondary aerosol formation due to abundant $NO_x$, $O_3$ and aerosols (e.g., Li et al., 2016; Tham et al., 2016; Wang et al., 2016; Wang et al., 2017a; Wang et al., 2017b; Yun et al., 2018). The $\gamma N_2O_5$ derived from ambient concentration measurements showed different characteristics and dependence compared to previous measurements in relatively clean environments (Wang et al., 2017b). To better understand the reactive uptake of $N_2O_5$ on complex ambient aerosols, a flow tube reactor approach was developed for direct $N_2O_5$ reactivity measurement under highly polluted conditions. In the following sections, we describe in detail the method used for determining the $N_2O_5$ uptake coefficient with an iterative box model, and discuss the factors affecting the system's performance and uncertainty. Laboratory tests and field deployment of the method are presented to demonstrate its application under conditions with high ambient concentrations of $NO_2/O_3$ and fresh NO emission.

## 2 Methodology

### 2.1 Flow tube reactor

The flow tube system consists of an $N_2O_5$ generation part, a sample inlet with aerosol filter manifold, a flow tube reactor and detection instruments. A schematic diagram of the experimental apparatus is given in Fig. 1. The sample inlet with an aerosol filter manifold is made of ¼-inch outer diameter (OD) stainless-steel tubing. By switching two stainless-steel ball valves, ambient air can be introduced directly into the flow tube or through a PTFE membrane (Pall Life Sciences) to remove aerosols. The flow tube is a Teflon-coated stainless-steel tube, 120 cm in length with an internal diameter of 12.5 cm. The ambient or filtered air enters and exits the flow tube via 10-cm-deep 60° tapered end caps. The total flow rate through the flow tube is 4.6 SLPM and includes 120 SCCM of $N_2O_5$ flow, which is introduced through an orthogonal entry to minimize the entrance length of the injected flow. The air pressure in the flow tube reactor is around 730 torr. The adopted flow rate and pressure give a Reynolds number of 55 (i.e., laminar flow) in the flow tube reactor. At the exit of the flow tube reactor, several detection instruments are used to measure the concentrations of $N_2O_5$, $O_3$, $NO_x$, and aerosol surface area.

**2.2 Generation of $N_2O_5$**

$N_2O_5$ is generated in-situ from the reaction of $O_3$ with excess $NO_2$ at room temperature via reactions (R1) and (R2), which has been used in many previous lab and field measurements (e.g., Bertram et al., 2009a).

$$O_3 + NO_2 \rightarrow NO_3 + O_2 \qquad\qquad (R1)$$

$$NO_3 + NO_2 + M \leftrightarrow N_2O_5 + M \qquad\qquad (R2)$$

In this study, ozone was generated from $O_2$ photolysis with a mercury lamp in a commercial calibrator (Model 4010, Sabio Instrument Inc.). A 100 SCCM of produced $O_3$ flow was mixed with 20 SCCM of $NO_2$ (10 ppmv balanced in $N_2$; Arkonic, USA) in a Teflon reaction chamber (volume = 68 $cm^3$) for about 28 s prior to injection into the flow tube reactor. Under the excessive $NO_2$ condition, the system was expected to shift the R2 equilibrium towards $N_2O_5$. Concentrations of synthesized $N_2O_5$ were calculated from observed changes in $NO_2$ (before and after addition of $O_3$), and the $N_2O_5$ content had also been inter-validated with a Cavity Ring Down Spectrometer (CRDS) in our previous studies (Wang et al., 2016). Prior to the $N_2O_5$ generation, the system was purged with dry zero air and $NO_2$ for at least two hours, to minimize the water content level and stabilize the $NO_2$ source. This system was shown to be able to produce $N_2O_5$ concentrations from 1 to 10 ppbv (after dilution in the flow tube). In typical experiment used in the present study, the input of the $N_2O_5$ source to the top of flow tube contained 4.3 ppbv of $N_2O_5$, together with 106 ppbv of $O_3$ and 57 ppbv of $NO_2$. The stability of synthetic $N_2O_5$ source was tested continuously for eight hours, and the variation of the signal was within ±2% in each hour. More detailed description of the $N_2O_5$ generation can be found in Wang et al. (2016).

**2.3 Detection instruments**

At the exit of the flow tube reactor, $O_3$ was measured by a UV photometric analyzer (Thermo, Model 49i) and $NO_2$ was measured by a chemiluminescence $NO_x$ analyzer (Thermo, Model 42i) equipped with a blue light photolytic converter (BLC). The aerosol number concentration and size distribution (10 nm to 10μm) were measured by a wide-range particle spectrometer (WPS, model 1000XP, MSP Corporation, USA) to determine the aerosol surface area. The uncertainty of the aerosol surface area measurement was 20-30% (Wang et al., 2017b; Tham et al., 2018). The transmission of aerosols in the flow tube was evaluated using laboratory-generated $(NH_4)_2SO_4$ particles. The passing efficiency was around 50% for particles with a size of 20 nm, and more than 90% for particles larger than 100 nm. The total surface area loss in the flow tube was around 10-25%. The $N_2O_5$ and $ClNO_2$ concentrations were quantified by an iodide-adduct chemical ionization mass spectrometer (CIMS; THS Instrument, Atlanta). The CIMS has been deployed in several field campaigns, and the setup and operation have been previously described (Tham et al., 2016; Wang et al., 2016; Wang et al., 2017a; Wang et al., 2017b). Briefly, the primary ion $I^-$ was generated from ionization of $CH_3I$ diluted in $N_2$ flow through a $^{210}Po$ source. The $N_2O_5$ and $ClNO_2$ were detected as ion clusters of $I(N_2O_5)^-$ and $I(ClNO_2)^-$ at 235 and 208 m/z by the quadrupole mass spectrometer. Because of the higher pipeline resistance in the flow tube reactor compared to ambient measurement, a smaller orifice with a 0.0135-in diameter was utilized

in the CIMS inlet to reduce the sample flow, and another orifice was added before the scroll pump to keep the pressure in the ionization reaction chamber at 50 torr. The corresponding sample flow was 0.4 SLPM. The detection limit of the instrument was estimated to be 2 pptv (1 min averaged data), and the uncertainty of the CIMS measurement was estimated as ±25% (Tham et al., 2016). The ambient VOCs were determined using an online gas chromatograph (GC) coupled with a flame ionization detector (FID) and a mass spectrometer (MS). The VOCs concentrations were used to determine the $k_{NO3-VOC}$ in the aerosol flow-tube system, which was treated as constant during the short-time period of flow tube measurement. The ambient NO level was measured by another chemiluminescence $NO_x$ analyzer (Thermo, Model 42i) equipped with a molybdenum converter.

## 3. Determination of residence time

The mean residence time that represents the average reaction time of the gases in the flow tube reactor is an essential parameter in calculation of the reactive uptake coefficient. In previous flow reactor studies (e.g., Thornton et al., 2003), the average residence time has usually been calculated from the flow rate and flow-tube volume assuming an ideal laminar flow. To determine the mean residence time for non-ideal flow more accurately, the Residence Time Distribution (RTD) method introduced by Danckwerts (1953) was used in the present study. The RTD method involves introduction of an inert tracer species into the reactor and detection of its transient concentration leaving the reactor outlet, and it has been widely used in previous lab studies to characterize the mixing and flow behavior of non-ideal aerosol flow reactors (e.g., Lambe et al., 2011).

Pulse injection of highly concentrated $ClNO_2$ was used in the present study to measure the RTD and hence determine the mean residence time. $ClNO_2$ is an inert gas within the dark Teflon-coated flow tube reactor and can be measured by CIMS with high time resolution (>1 Hz). $ClNO_2$ was synthesized in-situ via passing the $N_2O_5$ through a NaCl slurry in the Teflon tubing reactor (Wang et al., 2016). The pulse injection was controlled by a solenoid valve. At t=0 s, 120 SCCM (the same flow as $N_2O_5$ injection during the uptake measurement) of $ClNO_2$ was directly injected into the flow tube reactor; at t =2 s, the solenoid valve switched and the $ClNO_2$ flow was passed through a charcoal filter to provide zero gas into the flow-tube reactor. The RTD function E(t) is defined by the following equation:

$$E(t) = \frac{C_{(t)}}{\int_0^\infty C_{(t)}dt},$$

(3)

where the $C_{(t)}$ represents the $ClNO_2$ concentration measured at time t. Then the mean residence time can be calculated as follows:

$$\Delta t = \int_0^\infty tE(t)dt.$$

(4)

The measurement result of the residence time test is shown in Fig. 2. With a flow rate of 4.6 SLPM in the flow tube reactor, the mean residence time determined from the RTD method was 149±2 s. In comparison, the residence time calculated using

the flow rate and reactor volume gives a value of 159±5 s, which is 6.7% higher than that given by the RTD method, and could lead to underestimation of the rate constant. The RTD function in Fig. 2 is clearly different from the ideal laminar flow reactor. Bertram et al. (2009) have suggested that the determined rate constant would be underestimated by up to 25% due to non-ideal plug flow condition. More discussion of the uncertainty in $\gamma N_2O_5$ calculation associated with residence time distribution is presented in section 5.

## 4. Iterative box model for determination of loss rate and uptake coefficient

As described previously, the reactivity of $N_2O_5$ can be investigated using the aerosol modulation by comparing the loss rate of generated $N_2O_5$ in the flow tube reactor with and without ambient aerosols. Previous studies (e.g., Bertram et al., 2009a) utilized the exit-concentration ratio of $N_2O_5$ to obtain the $N_2O_5$ loss rate on aerosols. However, air mass changes lead to different $NO_3$ loss rates and production rates over a short time period (i.e., a typical sampling cycle for about 1 hr), and high background $NO_2$ and $O_3$ in the ambient air would affect the exit $N_2O_5$ concentration and hence bias the measurement of loss rate and uptake coefficient from the flow tube experiments. To minimize the potential influences of high ambient pollutants and rapidly changing air mass, a time-dependent box model constrained by the real measurement data was used in the present study to directly calculate the $N_2O_5$ loss rate in both aerosol and non-aerosol mode, considering multiple reactions describing the production and loss of $NO_3$ and $N_2O_5$ (R1–R6) in the ambient condition.

$$O_3 + NO \rightarrow NO_2 + O_2; \qquad\qquad k_3 \qquad\qquad\qquad (R3)$$

$$NO_3 + NO \rightarrow 2NO_2; \qquad\qquad k_{NO3-NO} \qquad\qquad (R4)$$

$$NO_3 + VOC \rightarrow products; \qquad\qquad k_{NO3-VOC} \qquad\qquad (R5)$$

$$N_2O_5 + aerosols/wall \rightarrow products, \qquad k_{het} = k_{wall} + k_{aerosols} \qquad\qquad (R6)$$

The rate constants for reactions R1 to R4 recommended by the National Aeronautics and Space Administration-Jet Propulsion Laboratory (Sander et al., 2009) were used. The loss rate coefficient $k_{NO3-VOC}$ from $NO_3$ reactions with VOCs (R5) was determined by ambient measured VOCs concentrations and rate coefficients from Atkinson and Arey (2003). The $N_2O_5$ heterogeneous loss rate coefficient $k_{het}$ (R6) including heterogeneous loss on both aerosol and reactor surfaces, was the only adjustable parameter while other parameters such as $N_2O_5$, NO, $NO_2$ and $O_3$ concentration were constrained by concurrent measurements. The model simulated the reactions starting from the entrance of the reactor after mixing the ambient air sample and synthetic $N_2O_5$ source. The initial concentrations of $[NO_2]_{t=0}$ and $[O_3]_{t=0}$ were calculated from the ambient measured levels of $NO_2$ and $O_3$ and those from $N_2O_5$ source. Given the constraint of measured parameters at the entrance of the flow tube reactor, including $[NO]_{t=0}$, $[NO_2]_{t=0}$, $[O_3]_{t=0}$, $[N_2O_5]_{t=0}$, $[VOCs]_{t=0}$, temperature and pressure, these reactions could be integrated in time (performed in Matlab with the Kinetic PreProcessor using a Radau5.integrator) (Damian et al., 2002) to obtain the exit concentrations of $NO_2$, $O_3$ and $N_2O_5$. The calculated concentrations were then compared with the measured concentrations at

the exit of the flow tube reactor, and the $N_2O_5$ loss rate coefficient was tuned until the $N_2O_5$ concentration predicted by the box model agreed with the measured $N_2O_5$ concentration, $[N_2O_5]_{\Delta t}$. Assuming that $k_{wall}$ are constant between successive flow tube experiments with and without aerosols, the loss rate coefficient on aerosols surfaces can be determined from the differences between two modes, $k_{aerosols} = k_{het}^{w/aerosols} - k_{het}^{wo/aerosols}$. Then the uptake coefficient of $N_2O_5$ on aerosol surfaces ($\gamma N_2O_5$) can be calculated by the following equation:

$$\gamma N_2O_5 = 4(k_{het}^{w/aerosols} - k_{het}^{wo/aerosols})/(c\ Sa) \tag{5}$$

In circumstances without concurrent ambient measurement of $NO_2$ and $O_3$ and when accurate measurements are only available at the flow tube outlet, as in the present study, an iterative box model including both backward and forward simulation is needed. Following the method suggested by Wagner et al. (2013), the relevant reactions can be integrated backward starting with the measured concentrations at the exit of the reactor ($t=\Delta t$) to obtain the initial concentrations. As the cycle between $NO_3$ and $N_2O_5$ is fast and quickly established in high $NO_x$ conditions, the $NO_3$ and $N_2O_5$ are considered as one singular $N_2O_5{*}$ species by assuming $NO_3$ and $N_2O_5$ are in equilibrium (Brown et al., 2003). Doing this also makes backward reaction simulation possible by avoiding unstable equilibrium in the box model. The NO at the entrance of the flow tube could react quickly with $O_3$ and $NO_3$, with a short lifetime of a few seconds, resulting in near zero concentration at the exit of the flow tube. To initialize the simulation, a time-dependent NO concentration in the flow tube must be derived. An approximate [NO] profile can be estimated from a forward simulation with inputs of measured initial NO, $N_2O_5$, guessed $k_{het}$ and estimated initial $NO_2$ and $O_3$ concentrations from the following equations. The measured initial NO data used three minutes earlier data as input data considering the mean residence time of 150 s.

$$[NO_2]_0 = [NO_2]_{\Delta t} \times e^{\Delta t\ k1[O_3]_{\Delta t}} - [NO]_0 \tag{6}$$

$$[O_3]_0 = [O_3]_{\Delta t} \times e^{\Delta t\ k1[NO_2]_{\Delta t}} + [NO]_0 \tag{7}$$

$$[NO]_t = [NO]_0 \times e^{-t\ (k3[O_3]_0 + \frac{k4[N_2O_5]_0}{Keq[NO_2]_0})} \tag{8}$$

The estimated [NO] profile was then constrained in the backward model simulation, together with inputs of measured concentrations of $N_2O_5$, $NO_2$, and $O_3$ at the exit of the flow tube reactor and the initially guessed $k_{het}$, to derive the initial mixing ratios. The box model was run forward and backward iteratively with updated values and adjusted $k_{het}$ until simulated $N_2O_5$ concentration matched the measurement at the exit of the flow tube reactor. The agreement of simulated $NO_2$ and $O_3$ concentrations with measurements was also used as a check to validate the model calculation. Thus, the uptake coefficient of $N_2O_5$ was determined from Eq. (5). An example of the iterative box model calculation is shown in Fig. 3.

For some conditions, the iterative box model returns a negative $N_2O_5$ loss rate coefficient. This non-physical result might result from much larger fluctuations of $k_{NO3}$ or $k_{wall}$ in the system during each measurement cycle. When $k_{aerosol}$ is small due to the

low $S_a$ or insignificant uptake, the $k_{NO3}$ or $k_{wall}$ may dominate the $N_2O_5$ loss in flow tube reactor, and the fluctuations of $k_{NO3}$ or $k_{wall}$ due to the air mass or temperature/RH changes would bias the $k_{aerosol}$ determination and led to large uncertainty or negative values. This situation often occurred under conditions of fresh NO emission; more discussion of the influence of NO is presented in section 6.

## 5. Laboratory test and overall uncertainty

Laboratory tests of $N_2O_5$ uptake on $(NH_4)_2SO_4$ aerosols were also performed with different NO, $NO_2$, and $O_3$ conditions, and the uptake coefficients were determined from the iterative box model analysis described above with input of measured concentrations. The determined uptake coefficient ranged from 0.018 to 0.026 (Table S1 in SI), which are similar to previous laboratory study results with $(NH_4)_2SO_4$ aerosols (Davis et al., 2008). The consistency also can serve as a validation of the applicability of the introduced system and method. In addition, we also compared the measured initial concentration of $NO_2$ and $O_3$ during the lab tests with that predicted from the iterative model (Fig. 3f). The $NO_2$ concentration matched well between model prediction and measurement, while $O_3$ showed a little lower from the model simulation, which might be due to the wall loss or other loss ways of $O_3$ in the flow tube reactor.

In the present work, the determination of $k_{aerosols}$ is independent of the magnitude of $k_{wall}$, but the stability of $k_{wall}$ is critical for the accurate retrieval of $k_{aerosols}$. $k_{wall}$ depends on RH, and the variability in RH on the time scale of the measurement can introduce additional uncertainty (Bertram et al., 2009a). Laboratory experimental tests have been conducted to investigate the variability of $k_{wall}$ with RH in the current flow tube system. $k_{wall}$ can be determined from the previously described iterative model with the measurement of $N_2O_5$ loss through the flow tube in a zero air flow in the absence of aerosols. As shown in Fig. 4, $k_{wall}$ has a strong positive relationship with RH, and increases with RH, especially when RH is higher than 50%. The consistent $k_{wall}$ at each RH condition with different initial $N_2O_5$ concentrations suggests that $k_{wall}$ in the current system is relatively stable under different chemical conditions but varies as a function of RH.

The sample air exiting the flow reactor was continuously measured by a RH probe, and the results showed that the RH variation between the aerosol presence and absence modes was within 1% more than 80% of the time during the ambient measurement cases. This result would translate into an uncertainty of ($\pm\ 0.15\times10^{-3}$) to ($\pm\ 2.4\times10^{-3}$) in $\gamma N_2O_5$ with RH of 20% to 70%, respectively and a Sa of 1000 $\mu m^2/cm^3$. To minimize the magnitude of the variability in $k_{wall}$, the wall of the reactor was coated with PFA-Teflon, and the flow tube reactor was cleaned daily with distilled water. Ultrasonic baths were also utilized after a one-week period of ambient measurement to remove aerosol build-up from the wall of the flow tube reactor.

In addition to $k_{wall}$ being affected by RH, uncertainty in $k_{aerosols}$ determination can also result from $N_2O_5$ source variability, $NO_3$ reactivity with VOCs, precision as well as accuracy associated with the measurement of all parameters. The long period of measurement cycle may also bring uncertainty due to concentrations variation in two operation modes. As described in Section

2.2, the stability of the $N_2O_5$ generation source was within ±2% over an hour. In the present study, online VOCs were measured with a time resolution of one hour. A ± 0.01 s$^{-1}$ variation of $k_{NO3-VOC}$ would lead to a single-point uncertainty in $\gamma N_2O_5$ of ± 0.4×10$^{-3}$ for Sa = 1000 µm$^2$/cm$^3$. NO reacts at a faster rate with $NO_3$, having a larger impact on the $\gamma N_2O_5$ calculation compared to VOCs. With a constrained real-time NO concentration, the iterative model can buffer against small NO changes. Stability

of NO, $NO_2$, $O_3$, and $N_2O_5$ for a period of at least 5 minutes for each mode is required to ensure that the flow-tube reactor measurement and iterative model yield reasonable results. The measurement precision and variation of these species during each cycle might also introduce uncertainty in the iterative model calculation.

The uncertainty in the $\gamma N_2O_5$ determination associated with $k_{wall}$ changes, VOCs variation, and the variation of the different parameters during the measurement cycles was estimated with a Monte Carlo approach, as described in Groß et al. (2014), by

assessing the uncertainty from individual key parameters (shown in Table 1) in the calculation model. $\gamma N_2O_5$ was found to be most sensitive to RH, which was closely related to $k_{wall}$ as discussed before. Fig. 5(a) shows the partial uncertainty of $\gamma N_2O_5$ derived from Monte Carlo simulations with RH at 40%. The single-point uncertainty in $\gamma N_2O_5$ was estimated to be ± 4.1×10$^{-3}$ for $\gamma N_2O_5$ around 0.03, and ± 3.6×10$^{-3}$ for $\gamma N_2O_5$ around 0.01, with RH of 40%. The uncertainty increased with RH and would be 9% to 17% at $\gamma N_2O_5$ around 0.03 for RH ranging from 20% to 70% (Fig. 5b).

Sensitivity tests with the iterative model calculation were performed to evaluate the uncertainty associated with measurement accuracy of $N_2O_5$ and VOCs, by varying the input $N_2O_5$ concentrations and $k_{NO3-VOC}$ in both modes. It is found that the $N_2O_5$ measurement uncertainty of 25% (Tham et al., 2016; Wang et al., 2017) would translate into an uncertainty of 12% in the $\gamma N_2O_5$ (shown in SI). The VOCs measurement uncertainty, however, has negligible influence on $\gamma N_2O_5$ calculation. In previous flow tube method introduced by Bertram et al., (2009), they also explained that the homogeneous reaction was expected to be

independent of the aerosol and non-aerosol modes and was thus can be canceled out in the calculation. Only strong atmospheric variation in VOC in short time period would influence the $N_2O_5$ uptake measurement. The uncertainty introduced by the aerosol surface area measurement including aerosol loss influence would be propagated to an uncertainty in the $\gamma N_2O_5$ calculation of 30%.

As mentioned in section 3, the use of mean residence time rather than RTD function by assuming an ideal reactor and ignoring

diffusion and dispersion processes would also introduce uncertainties. In order to evaluate the magnitude of this bias, we have performed a simplified test by comparing a first-order loss rate from mean residence time with a residence time distribution range. Briefly, the mean concentration of $N_2O_5$ at the exit the reactor could be expressed by:

$$\overline{[N_2O_5]} = \int_0^\infty [N_2O_5]_t E_t dt = \int_0^\infty [N_2O_5]_0 e^{-kt} E_t dt \qquad (9)$$

where [$N_2O_5$]t is the average concentration exit from the reactor between t and t + dt, E(t) is the residence time distribution

function, and k is the first order loss rate coefficient of $N_2O_5$. The results showed that the first-order loss rate calculated from

the distribution function was higher than that with a mean residence time, and was about 5% or 16% higher when the ratio of $\frac{[N_2O_5]_t}{[N_2O_5]_0}$ was 0.6 or 0.2 in the flow tube system, respectively. By incorporating all of these factors, the estimated total uncertainty is propagated to be 37% to 40% at $\gamma N_2O_5$ around 0.03 with 1000 $\mu m^2/cm^3$ Sa for RH ranging from 20% to 70%.

## 6. Demonstration of $\gamma N_2O_5$ measurements under polluted conditions

In polluted environments, high concentrations of $NO_2$, $O_3$ or NO in ambient air would affect the determination of the $N_2O_5$ loss rate and uptake coefficient in the flow tube experiments. To investigate the effect of multiple reactions of these species in polluted conditions, a series of tests with different conditions were simulated to compare the derived loss rate and uptake coefficient with and without consideration of $N_2O_5$ regeneration and NO titration in the flow tube system. Using the forward box model described in Section 4, the process in the flow tube reactor was simulated with an assumed fixed Sa of 1000 $\mu m^2/cm^3$, $\gamma N_2O_5$ of 0.03, $k_{wall}$ of 0.004 $s^{-1}$, and $k_{NO3\text{-}VOC}$ of 0.01 $s^{-1}$. Various conditions were simulated with different $O_3$, $NO_2$ and NO levels introduced into the flow tube, and the resulting concentrations of $N_2O_5$, $NO_2$, and $O_3$ at the exit of the reactors with and without aerosols modes were obtained. The loss rate and uptake coefficients of $N_2O_5$ were then calculated using the simple exit-concentration ratio approach (Eq. 2) and time-dependent iterative box model, respectively. The difference in $\gamma N_2O_5$ obtained from these two methods reflects the effect of $N_2O_5$ regeneration and NO titration on uptake coefficient determination.

Fig. 6 shows the simulation results for the derived uptake coefficients regarding the effect of $N_2O_5$ formation in the flow-tube reactor, with $O_3$ varied in the range of 0-100 ppbv and $NO_2$ in the range of 0-40 ppbv without NO presence in the ambient air. The $N_2O_5$ source input was fixed at 4.3 ppbv, as measured in the laboratory, together with 106 ppbv of $O_3$ and 57 ppbv of $NO_2$ from the $N_2O_5$ source. The $N_2O_5$ regeneration effect on $\gamma N_2O_5$ calculation was significant when $O_3$ and $NO_x$ levels in the ambient air were high. For example, at $NO_2 = 40$ ppbv and $O_3 = 100$ ppbv, which may frequently be encountered in city cluster regions in China, neglecting $N_2O_5$ formation in the flow tube would result in underestimating $\gamma N_2O_5$ by 42%.

To demonstrate the influence of NO titration, simulation tests were performed with NO varying from 0 to 8 ppbv. Because the reaction rate of NO with $NO_3$ is two orders of magnitude faster than that of NO with $O_3$, the initial $N_2O_5$ level would affect the NO titration process. We performed the simulation with different initial $N_2O_5$ concentrations injected into the flow-tube reactor. As the green line in Fig 7(a) indicates, the calculated $\gamma N_2O_5$ will be greatly underestimated when NO concentration increases, up to 55% at a NO level of 8 ppbv with an initial $N_2O_5$ level of 3 ppbv compared to NO level of zero. During the laboratory experiments, two initial $N_2O_5$ conditions with the input of additional 5 ppbv NO were also tested. The determined $\gamma N_2O_5$ from iterative model simulation and exit-concentration method was compared and shown as cubes in Fig. 7(a). The model results lie within the uncertainty range of the measurements, further cross-validating the NO influences and the model simulation. Fig. 7(a) also shows that a lower initial $N_2O_5$ leads to a larger underestimation of $\gamma N_2O_5$ in the presence of NO. It is not desirable

to use $N_2O_5$ concentrations above 5 ppbv to minimize the NO effect, because of other potential artifacts associated with working at high concentration (Thornton et al., 2003).

To explore which NO level would leave an extremely low $N_2O_5$ concentration in the exit of the reactor and make $N_2O_5$ loss rate measurement impossible, a series of experiments in clean air with additional NO was conducted in the laboratory to investigate NO titration effects and the performance of the iterative model in buffering against high NO. As shown in Fig 7(b), the derived $k_{het}$ showed consistent results for zero NO and NO < 6 ppbv conditions when RH and other parameters were unchanged. With higher NO addition and a lower initial $N_2O_5$ level, the calculated $k_{het}$, however, could be underestimated due to greater uncertainty when $NO_3$ and $N_2O_5$ were insufficient to titrate with NO. Fig. 7(b) also shows that the introduced box model method could buffer against NO below 8 ppbv with an initial $N_2O_5$ level of 4.3 ppbv. For future development, an activated-carbon scrubber in the inlet to reduce the gas-phase interferers (NO, $NO_2$, $O_3$, VOCs) but transmit aerosols could be a complementary approach to apply the flow tube system coupled with iterative box model analysis to even higher polluted conditions.

In summary, the simulation and laboratory results demonstrate that neglecting the formation and titration reactions in a flow tube reactor will result in underestimating $\gamma N_2O_5$. To reduce the NO titration effect, a relatively high level of $N_2O_5$ (but less than 5 ppbv) should be introduced to the flow tube reactor. Consideration of the multiple reactions in the iterative model is sufficiently robust to encourage further development to improve the accuracy of $\gamma N_2O_5$ calculations.

**7 Ambient measurement**

During winter 2017, the flow tube system was deployed to measure the $N_2O_5$ uptake coefficient at a sub-urban site in Heshan, Guangdong, in southern China. The sampling time for each mode with and without ambient aerosols lasted for at least 15 minutes to ensure 5 minutes' stable data at the exit for subsequent modeling analysis. The measured 5-min average concentrations of initial NO and exit $N_2O_5$, $NO_2$ and $O_3$ were used as the inputs in the iterative box model to derive $k_{het}$ and $\gamma N_2O_5$. Most measurements were conducted during the daytime to avoid interruption of nighttime ambient $N_2O_5$, and daytime $N_2O_5$ levels could be neglected. The average ambient temperature, RH, NO, $NO_2$, and $O_3$ during the field campaign were 23 ℃, 51%, 3.2 ppbv, 23 ppbv, and 32 ppbv respectively. As discussed previously, changes in RH and temperature can influence the stability of $k_{wall}$ and $N_2O_5$-$NO_3$ equilibrium, and thus upset $\gamma N_2O_5$ measurement. The cases where $\gamma N_2O_5$ measurement was affected by extreme fluctuations in NO (above 8 ppbv), temperature and RH (fluctuation >2%) were discarded from the analysis.

In addition to the iterative box model approach, we also used the exit-concentration ratio approach (c.f. Eq. 2) to calculate the $\gamma N_2O_5$. Fig.8 exhibits the comparison of $\gamma N_2O_5$ obtained using these two methods. Fifteen out of 51 measurements occurred under relatively "clean and stable" conditions (defined as ambient NO < 1 ppbv, fluctuation of NO < 0.3 ppbv, $NO_3$ production rate< 0.8 ppbv/min, and fluctuation of $NO_2$ and $O_3$ < 4 ppbv), and the corresponding values of $\gamma N_2O_5$ from the two methods

show good correlation, with an average ratio of 1.34, which is consistent with our previous simulation results that the exit-concentration ratio approach could underestimate $\gamma N_2O_5$ mainly due to $N_2O_5$ regeneration reaction. For conditions with higher precursor concentrations and fluctuations, the larger discrepancy between $\gamma N_2O_5$ from two methods was found (see Fig 8). As described previously, greater uncertainty in the exit-concentration ratio approach could result from multiple reactions and air mass changes. The fluctuations of NO, $NO_2$, and $O_3$ could greatly affect the exit $N_2O_5$ concentration ratio. For example, a lower NO level and higher $NO_2$, $O_3$ levels in the aerosol mode relative to the non-aerosol mode would result in a higher exit $N_2O_5$ concentration ratio, which would lead to underestimation of $\gamma N_2O_5$ and even negative values (see Fig.8 and SI). As even 1 ppbv fluctuation of NO concentration could largely affect exit $N_2O_5$ concentration, it would bring significant uncertainty to the exit-concentration ratio approach. When NO concentration is much higher, for example in the aerosol existing mode, the measured $N_2O_5$ concentration would be lower due to NO titration, thus overestimate the uptake coefficient if only comparing the end concentration ratio of $N_2O_5$ in two modes.

Two example cases with large air mass changes are shown in Fig. 9. In Fig. 9(a), a case with high and fluctuating NO emission was observed on the night of March 21, 2017, with average ambient concentrations of NO of 6 ppbv, $NO_2$ of 27 ppbv, $O_3$ of 2 ppbv, and Sa of 1880 $\mu m^2/cm^3$. $\gamma N_2O_5$ was determined to be 0.028 from the iterative model approach, and a higher $\gamma N_2O_5$ value of 0.036 was obtained from the exit-concentration ratio approach. The overestimated $\gamma N_2O_5$ from the exit-concentration ratio approach could be explained by the increased NO level (~1.5 ppbv) in the aerosol mode. For comparison, another two periods of data points in the March 21 case (Fig. 9a) with different NO levels were also selected to derive the $k_{het}$, and the results showed good consistency (0.0136-0.0140 $s^{-1}$) (Fig S2 in SI), also demonstrating the applicability of the iterative model in buffering against fluctuated NO. In Fig. 9(b), another case with fluctuating $NO_2$ and $O_3$ levels was observed on March 26, 2017, and the $NO_2$ level was about 5 ppbv higher but the $O_3$ level was about 11 ppbv lower in aerosol mode. With Sa of 681 $\mu m^2/cm^3$, $\gamma N_2O_5$ was determined to be 0.020 from the iterative model approach and a much lower value of 0.008 from the exit-concentration ratio approach. The consideration of multiple reactions in the iterative model approach was able to buffer against small fluctuations of precursors in switching between aerosol and non-aerosol modes. The results demonstrated the applicability of the iterative model approach to directly measuring the $N_2O_5$ heterogeneous uptake coefficient under conditions of high $NO_2/O_3$ and fresh NO emission.

**8 Summary and conclusion**

An in-situ experimental approach for direct measurement of $N_2O_5$ heterogeneous reactivity in a polluted environment was developed and introduced in the present study. The method uses an aerosol flow tube reactor combined with an iterative box model, to determine the heterogeneous loss rate of synthesized $N_2O_5$ on ambient aerosols with consideration of multiple reactions affecting $N_2O_5$ in the flow tube. A series of laboratory and model simulations were conducted to test the applicability of the system with different conditions. The overall $\gamma N_2O_5$ uncertainty from the variations of parameters during two operation

modes and uncertainties associated with measurements of gaseous and aerosol species was propagated to be 37-40% at $\gamma N_2O_5$ around 0.03 with Sa of 1000 $\mu m^2/cm^3$ and RH ranging from 20% to 70%. Field deployment of this system at a polluted suburban site in South China demonstrated the applicability of the introduced method in measuring $N_2O_5$ uptake coefficients in polluted environments with high ambient levels of $O_3$, NO and $NO_2$ and rapid air mass changes. Both field results and simulation tests demonstrate that neglecting multiple reactions within the flow tube reactor leads to underestimating $\gamma N_2O_5$ values. The introduced approach could also be used to investigate the heterogeneous reactivity of other trace gases on ambient aerosols in polluted environments.

**Acknowledgment**

This work was funded by the National Natural Science Foundation of China (91544213, 41505103), the Research Grants Council of Hong Kong Special Administrative Region, China (C5022-14G, 15265516) and the National Key R&D Program of China (No. 2016YFC0200500). The authors also acknowledge the support of the Research Institute for Sustainable Urban Development (RISUD).

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

**Table 1: Parameters varied in the Monte-Carlo simulations**

| Parameter | Fixed value | Variation [a] | Parameter | Fixed value | Variation [a] |
|---|---|---|---|---|---|
| $[NO_2]_t$ | 53.5 ppbv | 0.3 ppbv | $[N_2O_5]_0$ | 5.0 ppbv | 0.1 ppbv |
| $[NO]_0$ | 2 ppbv | 0.1 ppbv | $[N_2O_5]_t$ | 1.8 ppbv | 0.1 ppbv |
| $[O_3]_t$ | 78.8 ppbv | 0.6 ppbv | $k_{NO3-VOC}$ | 0.01 s$^{-1}$ | 0.01 s$^{-1}$ |
| Temperature | 25 ℃ | 0.1℃ | RH | 20-70% | 1% |
| Residence Time | 150 s | 2 s | $k_{wall}$ | * | ** |

[a] $1\sigma$ standard deviation for the varied parameters.

* The $k_{wall}$ is calculated from RH, using the relation fitting equation in Fig 4.

** The variation of $k_{wall}$ is calculated as RH varied 1%.

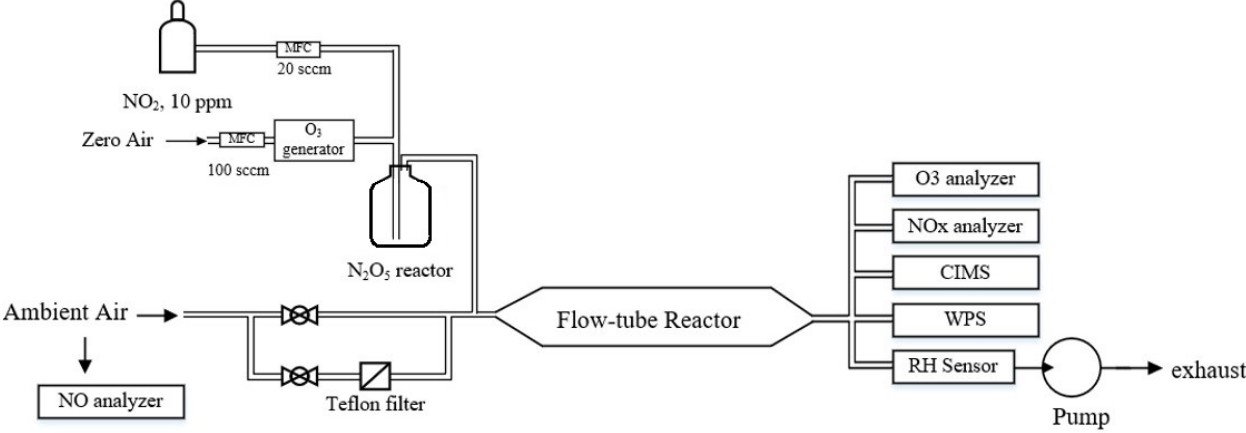

**Figure 1: Schematic diagram of the aerosol flow tube system.**

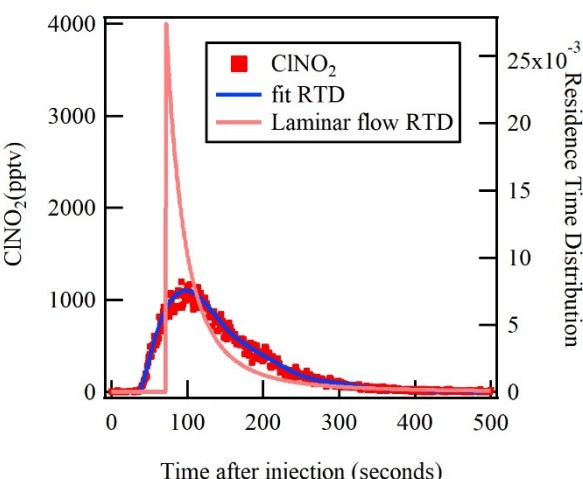

**Figure 2: The measured residence time distribution of the injected ClNO₂ in the flow-tube reactor. The blue line represents the fitted residence time distribution of the ClNO₂ pulse injection experiment. The pink line represents the expected residence time distribution of an ideal laminar flow reactor without diffusion.**

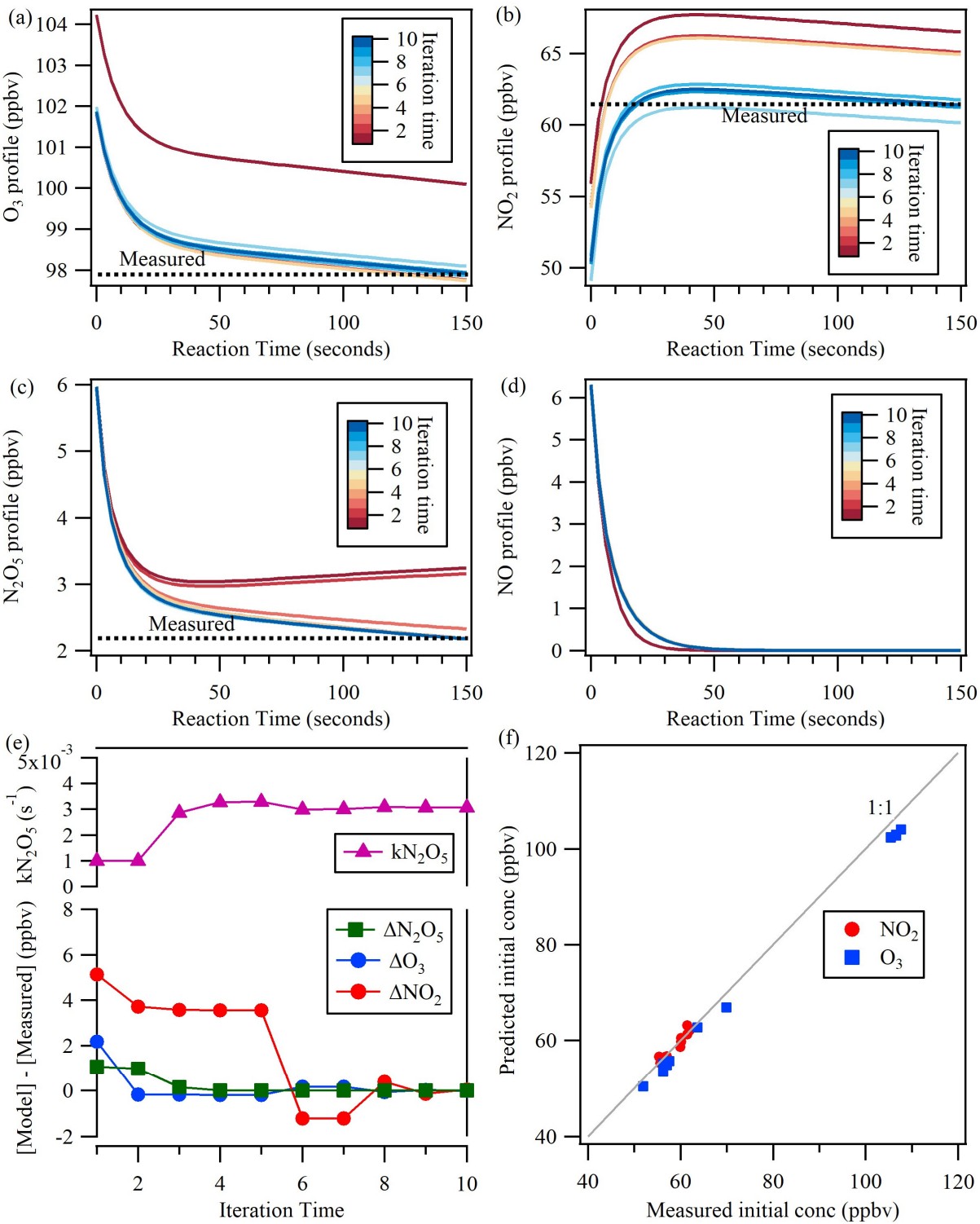

**Figure 3: An example of the iterative box model simulation to derive $k_{het}$ from the measured concentrations of $NO_2$, $O_3$ and $N_2O_5$ at the exit of the flow tube reactor. The concentration profiles obtained from the simulation in 10 iterations are shown for (a) $O_3$, (b) $NO_2$, (c) $N_2O_5$, and (d) NO. In the upper panel of (e), the adjusted $N_2O_5$ loss rate is shown for each iteration. The lower panel of (e) shows the concentration differences between the model simulation and measurements of $N_2O_5$, $O_3$ and $NO_2$ at the exit of the reactor for each iteration. Panel (f) shows the comparison between measured initial concentrations from laboratory test and predicted initial concentrations from the iterative model.**

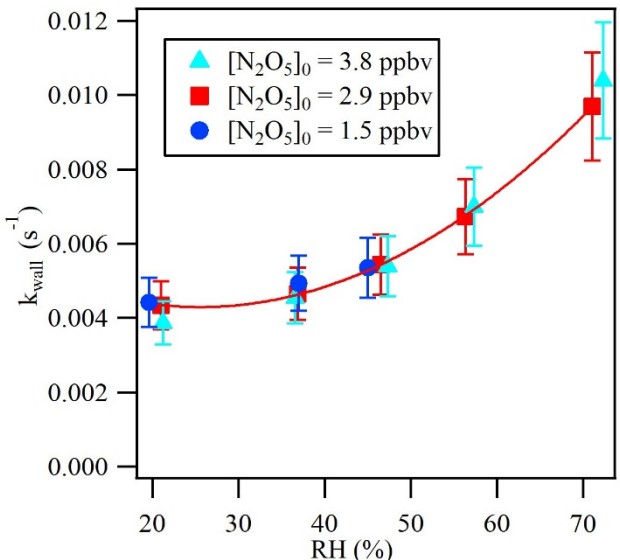

**Figure 4: Relative humidity dependence of the wall loss rate coefficient ($k_{wall}$) of $N_2O_5$ in the flow reactor.**

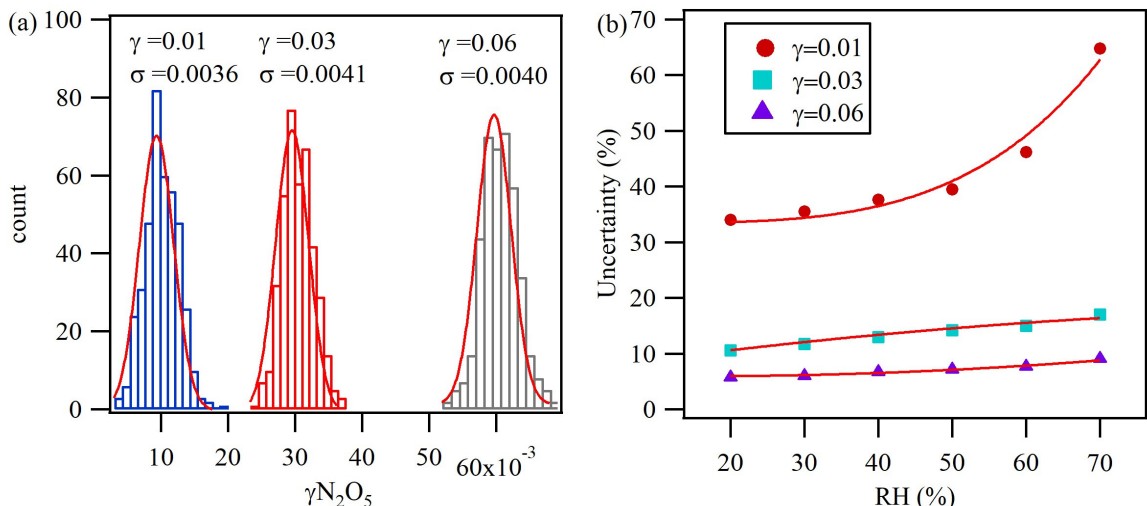

**Figure 5: The partial uncertainty in γN₂O₅ determination associated with k_wall changes, VOCs variation, and the variation of different parameters during the measurement cycles derived from Monte Carlo simulations for three individual sets with 400 simulations at (a) RH = 40% and (b) different RH values. In these three data sets, the condition was set as following: surface area=1000 μm²/cm³, reaction time = 150 s, initial O₃ = 80 ppbv, initial NO₂ = 50 ppbv, initial NO = 2 ppbv, initial N₂O₅ = 5 ppbv, temp = 25℃, k_NO3-VOC = 0.01 s⁻¹.**

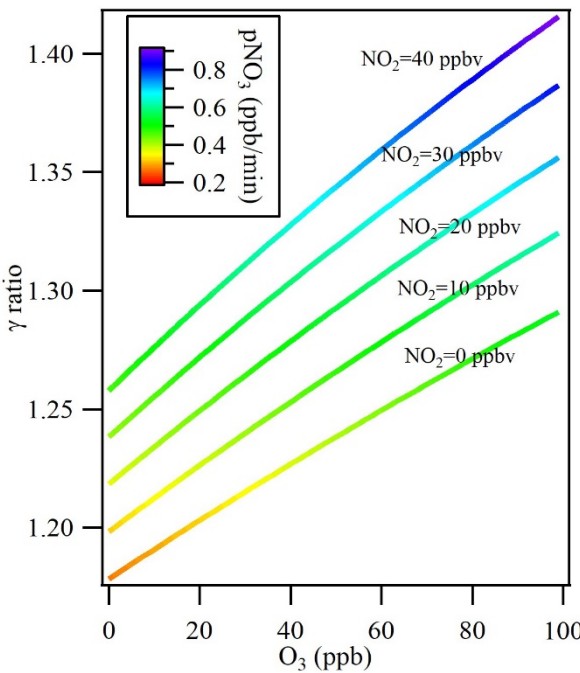

**Figure 6: The influence of multiple reactions resulting from high ambient NO₂ and O₃ levels under different ambient NO₂ levels from 0-40 ppbv. The colors indicate the NO₃ production rate (pNO₃) at the entrance of the flow tube reactor after mixing with 106 ppbv of O₃ and 57 ppbv of NO₂ from the N₂O₅ source.**

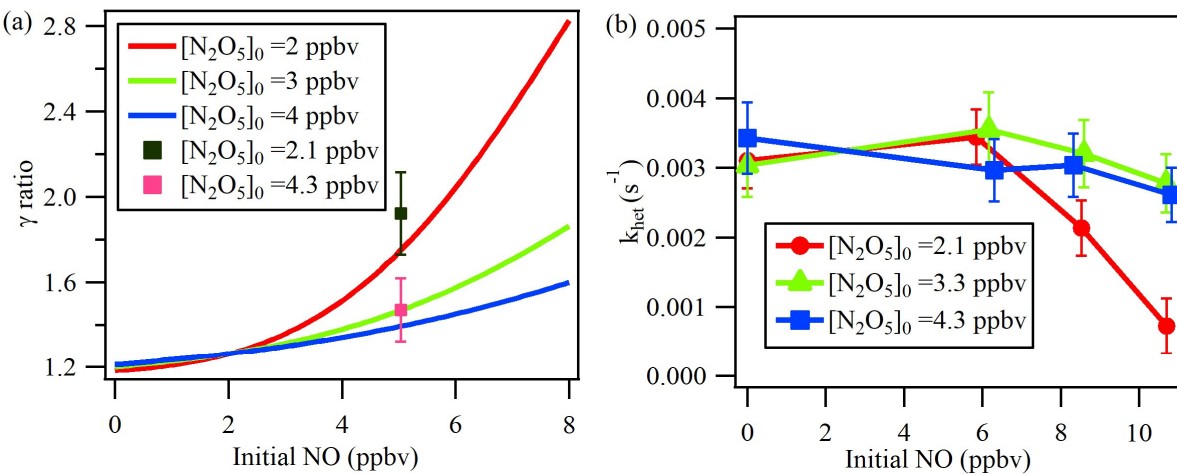

**Figure 7: (a) Simulation results of NO titration effect on γN₂O₅. The γN₂O₅ ratio represents (γN₂O₅ from the iterative model) / (γN₂O₅ from ignoring multiple reactions method). Initial NO and initial N₂O₅ represent the respective initial concentrations of NO and N₂O₅ in the flow tube reactor. The lines represent the simulation result and the cubes represent the lab test result. (b) k_het calculated via the iterative model in laboratory experiments with constant RH of 21%, different initial N₂O₅, and varied NO additions.**

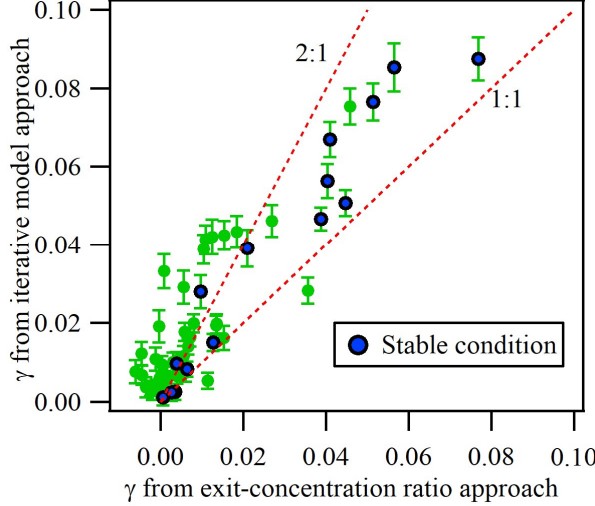

**Figure 8: Comparison of γN₂O₅ determined from the exit-concentration ratio approach and the iterative model approach for all available data measured in the Heshan campaign. The blue points represent the data obtained under "clean and stable condition", while green points are data obtained from other condition. The "clean and stable condition" is defined as follows: ambient NO < 1 ppbv, the change of NO < 0.3 ppbv, the NO₃ production rate < 0.8 ppbv/min, and the change of NO₂ and O₃ < 4 ppbv. The error bar represents the uncertainty calculated by Monte Carlo approach under the measurement condition.**

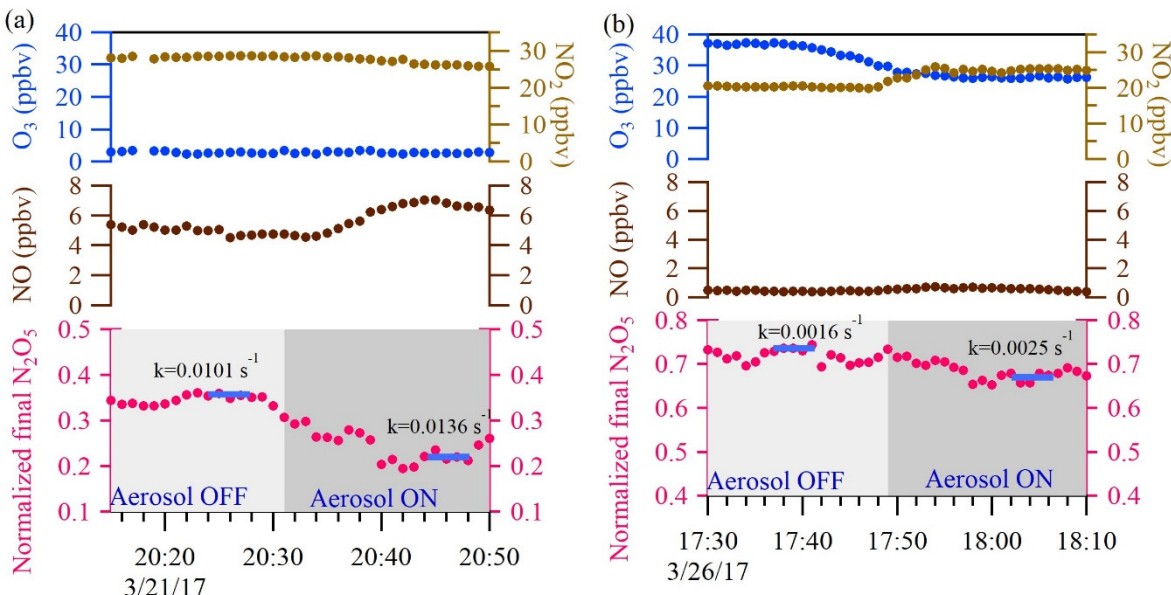

**Figure 9: Two sample cases are shown. In the upper panel, the blue and light brown dots represent 1-min ambient $O_3$ and $NO_2$ data, respectively. In the middle panel, the brown dots represent 1-min ambient NO data. In the lower panel, the pink dots represent 1-min average of $N_2O_5$ concentration normalized to the initial $N_2O_5$ concentration in the flow-tube reactor. The calculated total $N_2O_5$ loss rate derived from the iterative model with 5-min average input data (the blue bar) is also shown for each cycle.**