# Peer review of "An in-situ flow-tube system for direct measurement of N2O5 heterogeneous uptake coefficients in polluted environments"

_Atmospheric Measurement Techniques, 2018_

## Referee Comment (RC1) · Anonymous Referee #1 · 17 Jul 2018

Review of "An in-situ flow-tube system for direct measurement of N2O5 heterogeneous uptake coefficients in polluted environments"

The authors present a flow tube measurement of the N2O5 uptake coefficient that is an extension of the work of Bertram, Riedel, and Thornton. The measurement system is described and it is similar to the earlier design. The main innovations presented here are a more detailed measurement of the residence time distribution in the flow tube and the application of an iterative box model to retrieve the uptake coefficient when ambient concentrations of NO, NO2 and O3 are high enough to make 2nd order reactions important in the flow tube. The authors also present ambient measurements of the uptake coefficient which are useful because these direct measurements are rare and limited geographically.

This is an important measurement and should be published in AMT with minor changes.

Suggestion: A method, complementary to the iterative box model analysis, would be to reduce the concentrations of the gas-phase interferers (NO, NO2, O3, VOCs) before the N2O5 addition using an actived-carbon scrubber that transmits aerosols, such as http://www.sunlab.com/denuders/ .

Minor issues:

1) Typically laboratory measurements of the uptake coefficient on synthetic aerosol are less than 0.04. Although some ambient analyses (Wagner et al. 2013, McDuffie et al. 2018) report uptake coefficients above 0.04 (upto 0.1) for a small subset of the data. It is not clear if these are artifacts of the analysis or real measurements of the uptake coefficient. Here the authors also report a direct measurements of uptake coefficients between 0.04 and 0.1. I would encourage the authors to address the discrepancy between laboratory measurement and their ambient measurements.

   If they are real what is aerosol composition? Can the measured uptake be reproduced in the lab with synthetic aerosol?

2) It is unclear what parameters were used in the uncertainty analysis. I suspect uncertainty due to the aerosol surface area measurement would be at least +/-25%. In figure 9, there are not smooth exponential decay transitions between filter ON and OFF periods,so I suspect the uncertainty in the N2O5 measurement is significant.

   On page 8 line 18,please list the key parameters and the uncertainty associated with them.

3) Measurements of NO and VOCs are not described. Uncertainty due to reactions of NO3 with unmeasured VOCs should be bounded.

4) The authors show that the residence time in the flow tube is a distribution (ranging over a factor of 2 in residence times), however in the iterative box model only the mean residence time is used. As the iterative box model likely depends in the residence time in a nonlinear way, the author should use a range of residence times in the iterative box model.

5) Have the authors measured particle losses in the flow tube? Diffusional and gravitational losses could be important. Could aerosol losses also be RH dependent? If so, please add a few sentences describing the results.

6) In figure 9, the periods chosen for analysis seems to be handpicked for stability. If different periods were chosen how would the results change?

Technical issues:
Pg 3, line 26: How does the flow tube pressure relate to ambient pressure?
Pg 4, line 14: how much $NO_2$ is added with the $N_2O_5$ addition?
Pg 7, line 4: This sentence is missing a subject
Pg 7, line 20: Please give some more explanation about when non-physical results occur. When the uptake coefficient is small. When aerosol number is low?  I expect that in a flow tube with high initial $N_2O_5$ the box model would work well in most cases.
Pg 8, line 5: please add 'respectively'
Pg 9 line 9: typo 'ere'
Pg 9, line 16: Could you summarize the potential artifacts.
Pg. 10 line 27: missing 'the', 'in aerosol mode'

---

## Referee Comment (RC2) · Anonymous Referee #2 · 2 Aug 2018

General Comments The authors propose a new variation of the N2O5 reactivity measurement introduced by Bertram et al in 2009. Specifically, the authors utilize an iterative box model coupled with measurements of NO, NO2, and O3 to compute the loss rate of N2O5 in the flow reactor when high and variable concentrations of NO, NO2, and O3 complicate the retrieval of N2O5 uptake coefficients. The paper is suitable for publication following the authors attention to the comments below:

1) I strongly encourage the authors to show results of laboratory tests on a model aerosol (e.g., NaCl or (NH4)2SO4) with varying inlet concentrations of NO, NO2, and O3 as this will cement the uncertainty analysis and the retrieval of N2O5 uptake co-

[Figure]

efficients that are reported here. 2) Often, NO3 reactivity can be dominated by VOCs (e.g., isoprene)? If these VOCs are not measured, their effects on N2O5 uptake would not be captured by the model. Discussion of the potential effects should be included.

Specific Comments: Page 2 Line 4: The units do not cancel when representing C in m/s and Sa in um2/cm3. Either remove the units or place all in common units m2/m3 for surface area.

Page 2 Line 9: What is a "pure" or "synthetic" aerosol? I would replace with model aerosol compounds based on the references cited.

Page 2 Line 27: The flow tube of Bertram et al was deployed to sites in Boulder, CO and Seattle, WA, and La Jolla, CA. I would not characterize any of these sites as rural, based on local NOx concentrations.

Page 4 Section 2.2: What is the concentration of NO2 and O3 in the flow tube?

Page 4 Section 2.3: Please confirm that surface area was measured at same RH of the flow tube. Also, was RH measured in the flow tube?

Section 3: The RTD by definition is a distribution of residence times. The shape of this distribution can bias the retrieved N2O5 uptake coefficients. If the distribution is normal, I would expect use of the mean residence time to be appropriate. If the distribution is not normally distributed, then the tails of the distribution can impact the retrieval of the N2O5 uptake coefficient. The authors site a mean of 149 +/-2, but that does not capture the distribution in residence time. Error induced by having a distribution of reaction times should be discussed in more detail here. I expect that this factor alone will carry uncertainty that is larger than the 9-17% cited in the abstract.

Section 5: The propagation of errors and calculation of the overall uncertainty from the Monte Carlo method is interesting. It should be clearly stated that the uncertainty is a strong function of Sa. The number cited are for 1000 um2/cm3, for delta RH (aerosol on vs off) of less than 1% and for a specific delta in NO3 reactivity (0.01 s-1, between

aerosol on and off). This should be cast in terms of an equivalent [NO].

Page 9 Line 11: The retrieval of the N2O5 uptake coefficient is sensitive to a difference in NO3 reactivity between the aerosol on and off states. It would be helpful if the authors also stated how the difference in NO concentration between the on and off states impacted the retrieval.

––––––––––––––––––––––––––––––

---

## Author Comment (AC2) · 14 Sep 2018

**Response to Anonymous Referee #2**

**General Comments**

The authors propose a new variation of the N2O5 reactivity measurement introduced by Bertram et al in 2009. Specifically, the authors utilize an iterative box model coupled with measurements of NO, NO2, and O3 to compute the loss rate of N2O5 in the flow reactor when high and variable concentrations of NO, NO2, and O3 complicate the retrieval of N2O5 uptake coefficients. The paper is suitable for publication following the authors attention to the comments below:

**Response**: We thank the reviewer for his/her attention to this manuscript. We have made all the suggested changes and/or made clarifications. The reviewer's comment is in black and our response is in blue wording and the revised text is in italic.

1) I strongly encourage the authors to show results of laboratory tests on a model aerosol (e.g., NaCl or (NH4)2SO4) with varying inlet concentrations of NO, NO2, and

O3 as this will cement the uncertainty analysis and the retrieval of N2O5 uptake coefficients that are reported here.

**Response**: Thanks for the valuable suggestions. The results of laboratory tests with  $(NH_4)_2SO_4$  aerosols with the same system is now included in the revised text and SI.

The revised text reads,

"Laboratory tests of  $N_2O_5$  uptake on  $(NH_4)_2SO_4$  aerosols were also performed with different NO, NO2, and O3 conditions, and the uptake coefficients were determined from the iterative box model analysis described above with input of measured concentrations. The determined uptake coefficient ranged from 0.018 to 0.026 (Table S1 in SI), which are similar to previous laboratory study results with  $(NH_4)_2SO_4$  aerosols (Davis et al., 2008). The consistency also can serve as a validation of the applicability of the introduced system and method. In addition, we also compared the measured initial concentration of NO2 and O3 during the lab tests with that predicted from the iterative model (Fig 3f). The NO2 concentration matched well between model prediction and measurement, while O3 showed a little lower from the model simulation, which might be due to the wall loss or other loss ways of O3 in the flow tube reactor."

"During the laboratory experiments, two initial  $N_2O_5$  conditions with the input of additional 5 ppbv NO were also tested. The determined  $\gamma N_2O_5$  from iterative model simulation and exitconcentration method was compared and shown as cubes in Fig 7(a). The model results lie within the uncertainty range of the measurements, further cross-validating the NO influences and the model simulation."

The lab experiment conditions and derived uptake coefficients are also listed in Table S1 in SI.

| No. | Initial NO 2
(ppb) | Initial O 3
(ppb) | Initial NO
(ppb) | Initial N 2 O 5
(ppb) | RH (%) | Sa (µm 2 /cm 3 ) | γ      |
|-----|----------------------------------|---------------------------------|---------------------|------------------------------------------------|--------|----------------------------------------|--------|
| 1   | 62                               | 57                              | 0                   | 2.1                                            | 25.1   | 848                                    | 0.0226 |
| 2   | 62                               | 57                              | 5.0                 | 2.1                                            | 24.6   | 928                                    | 0.0208 |
| 3   | 57                               | 106                             | 0                   | 4.3                                            | 22.9   | 965                                    | 0.0182 |
| 4   | 57                               | 106                             | 5.0                 | 4.3                                            | 23.2   | 894                                    | 0.0212 |
| 5   | 57                               | 106                             | 0                   | 4.3                                            | 48     | 1425                                   | 0.0259 |

Table S1. Lab experiments with (NH4)2SO4 aerosols.

2) Often, NO3 reactivity can be dominated by VOCs (e.g., isoprene)? If these VOCs are not measured, their effects on N2O5 uptake would not be captured by the model. Discussion of the potential effects should be included.

**Response:** Yes, the gas-phase reactions between NO3 and VOCs can affect the N2O5 reactivity measurement. In the both flow tube methods introduced by Bertram et al. (2009) and that in the present study, the homogeneous reaction is expected to be independent of the aerosol and non-aerosol modes and is thus can be cancelled out in the calculation. Only strong atmospheric variation in VOC in short time period will influence the N2O5 uptake measurement. In the present study, VOCs including isoprene and monoterpenes were measured by an online-GC with time-resolution of 1 hour. Thus, the kNO3-VOC in the aerosol flow-tube system was treated as constant during each measurement cycle. The uncertainty from kNO3-VOC variation is addressed by Monte Carlo approach and is found that  $\pm$  0.01 s-1 variation of kNO3-VOC would lead to a single-point uncertainty in  $\gamma$ N2O5 of  $\pm$  0.4×10-3 for Sa = 1000 µm2/cm3. In addition, we have also run a sensitivity test with half or doubled kNO3-VOC as input value in the model, to address the effect of uncertainty in VOCs measurement, the results show that the effect of VOCs uncertainty was negligible. More information of the VOC measurements and more discussion on the potential influences are added in the revised text, as follows,

"Sensitivity tests with the iterative model calculation were performed to evaluate the uncertainty associated with measurement accuracy of  $N_2O_5$  and VOCs, by varying the input  $N_2O_5$  concentrations and  $k_{NO3-VOC}$  in both modes. It is found that the  $N_2O_5$  measurement uncertainty of 25% (Tham et al., 2016; Wang et al., 2017) would translate into an uncertainty of 12% in the  $\gamma N_2O_5$  (shown in SI). The VOCs measurement uncertainty, however, has negligible influence on  $\gamma N_2O_5$  calculation. In previous flow tube method introduced by Bertram et al., (2009), they also explained that the homogeneous reaction was expected to be independent of the aerosol and non-aerosol modes and was thus can be cancelled out in the calculation. Only strong atmospheric variation in VOC in short time period would influence the  $N_2O_5$  uptake measurement."

Figure S1. Sensitivity test of iterative model via varying input N2O5 and kNO3-VOC in both modes.

Specific Comments:

Page 2 Line 4: The units do not cancel when representing C in m/s and Sa in um2/cm3. Either remove the units or place all in common units m2/m3 for surface area.

**Response:** The unit of surface area is corrected as  $m^2/m^3$ .

"where  $c_{N2O5}$  (m/s) is the mean molecular speed of  $N_2O_5$  and Sa ( $m^2/m^3$ ) is the aerosol surface area concentration."

Page 2 Line 9: What is a "pure" or "synthetic" aerosol? I would replace with model aerosol compounds based on the references cited.

**Response:** The sentence has been changed to:

"in the presence of pure inorganic and organic aerosols or mixed aerosols under different conditions"

Page 2 Line 27: The flow tube of Bertram et al was deployed to sites in Boulder, CO and Seattle, WA, and La Jolla, CA. I would not characterize any of these sites as rural, based on local NOx concentrations.

**Response:** Thanks for pointing this out. We have corrected the description to 'urban sites', as follows:

"This flow tube apparatus was deployed at two urban sites in Boulder and one coastal site in La Jolla to measure  $\gamma N_2 O_5$  on ambient aerosols (Bertram et al., 2009b; Riedel et al., 2012)."

Page 4 Section 2.2: What is the concentration of NO2 and O3 in the flow tube?

**Response:** With the dilution of zero air, the concentration of  $NO_2$  and  $O_3$  was round 57 and 106 ppb at the top of the flow tube reactor. This information has been added in the revised text.

"In typical experiment used in the present study, the input of the  $N_2O_5$  source to the top of flow tube contained 4.3 ppbv of  $N_2O_5$ , together with 106 ppbv of  $O_3$  and 57 ppbv of  $NO_2$ ."

Page 4 Section 2.3: Please confirm that surface area was measured at same RH of the flow tube. Also, was RH measured in the flow tube?

**Response:** Yes, the surface area was measured at same RH of the flow tube, because we didn't add any aerosol drier before the WPS when doing the flow tube measurement. The RH was continuously measured at the exit of the flow tube reactor, as shown in Fig. 1.

Section 3: The RTD by definition is a distribution of residence times. The shape of this distribution can bias the retrieved N2O5 uptake coefficients. If the distribution is normal, I would expect use of the mean residence time to be appropriate. If the distribution is not normally distributed, then the tails of the distribution can impact the retrieval of the N2O5 uptake coefficient. The authors site a mean of  $149 \pm 2$ , but that does not capture the distribution in residence time. Error induced by having a distribution of reaction times should be discussed in more detail here. I expect that this factor alone will carry uncertainty that is larger than the 9-17% cited in the abstract.

**Response:** Thanks for the valuable suggestion. We agree with the reviewer that using the mean residence time could bring large errors into the uptake coefficient determination. Because it is very difficult to include the RTD function in the iterative model calculation, we have performed a simplified test to estimate the uncertainty that may arise from the use of mean residence time. As also stated in the response to reviewer #1, we have added more information and more discussion in the revised text, as follows,

"The RTD function in Fig. 2 is clearly different from the ideal laminar flow reactor. Bertram et al. (2009) have suggested that the determined rate constant would be underestimated by up to 25% due to non-ideal plug flow condition. More discussion of the uncertainty in  $\gamma N_2 O_5$  calculation associated with residence time distribution is presented in section 5."

"As mentioned in section 3, the use of mean residence time rather than RTD function by assuming an ideal reactor and ignoring diffusion and dispersion processes would also introduce uncertainties. In order to evaluate the magnitude of this bias, we have performed a simplified test by comparing a first-order loss rate from mean residence time with a residence time distribution range. Briefly, the mean concentration of  $N_2O_5$  at the exit the reactor could be expressed by:

$$\left[\overline{N_2 O_5}\right] = \int_0^\infty [N_2 O_5]_t E_t dt = \int_0^\infty [N_2 O_5]_0 e^{-kt} E_t dt \tag{9}$$

where  $[N_2O_5]t$  is the average concentration exit from the reactor between t and t + dt, E(t) is the residence time distribution function, and k is the first order loss rate coefficient of  $N_2O_5$ . The results showed that the first-order loss rate calculated from the distribution function was higher than that with a mean residence time, and was about 5% or 16% higher when the ratio of  $\frac{[N_2O_5]t}{[N_2O_5]_0}$  was 0.6 or 0.2 in the flow tube system, respectively."

Section 5: The propagation of errors and calculation of the overall uncertainty from the Monte Carlo method is interesting. It should be clearly stated that the uncertainty is a strong function of Sa. The number cited are for 1000 um2/cm3, for delta RH (aerosol on vs off) of less than 1% and for a specific delta in NO3 reactivity (0.01 s-1, between aerosol on and off). This should be cast in terms of an equivalent [NO].

**Response:** According to our and other previous studies, the uncertainty of the aerosol surface area measurement from the WPS system could be around 20-30% (Wang et al., 2017; Tham et al., 2018). The Monte Carlo simulation was only used to consider the  $k_{wall}$  changes, VOCs variation, and the variation of the different parameters during two modes in the measurement cycle. In addition, sensitivity tests were also included and the overall uncertainty by incorporating all of the factors are now updated. The revised text is as follows,

"The uncertainty of the aerosol surface area measurement was 20-30% (Wang et al., 2017; Tham et al., 2018)."

"In addition to  $k_{wall}$  being affected by RH, uncertainty in  $k_{aerosols}$  determination can also result from  $N_2O_5$  source variability,  $NO_3$  reactivity with VOCs, precision as well as accuracy associated with the measurement of all parameters. The long period of measurement cycle may also bring uncertainty due to concentrations variation in two operation modes. As described in Section 2.2, the stability of the  $N_2O_5$  generation source was within  $\pm 2\%$  over an hour. In the present study, online VOCs were measured with a time resolution of one hour.  $A \pm 0.01 \text{ s}^{-1}$ variation of  $k_{NO3-VOC}$  would lead to a single-point uncertainty in  $\gamma N_2O_5$  of  $\pm 0.4 \times 10^{-3}$  for Sa =  $1000 \ \mu m^2/cm^3$ . NO reacts at a faster rate with NO3, having a larger impact on the  $\gamma N_2O_5$ calculation compared to VOCs. With a constrained real-time NO concentration, the iterative model can buffer against small NO changes. Stability of NO, NO2, O3, and  $N_2O_5$  for a period of at least 5 minutes for each mode is required to ensure that the flow-tube reactor measurement and iterative model yield reasonable results. The measurement precision and variation of these species during each cycle might also introduce uncertainty in the iterative model calculation.

The uncertainty in the  $\gamma N_2O_5$  determination associated with  $k_{wall}$  changes, VOCs variation, and the variation of the different parameters during the measurement cycle was estimated with a Monte Carlo approach, as described in Groß et al. (2014), by assessing the uncertainty from individual key parameters (shown in Table 1) in the calculation model.  $\gamma N_2O_5$  was found to be most sensitive to RH, which was closely related to  $k_{wall}$  as discussed before. Fig. 5a shows the partial uncertainty of  $\gamma N_2O_5$  derived from Monte Carlo simulations with RH at 40%. The singlepoint uncertainty in  $\gamma N_2O_5$  was estimated to be  $\pm 4.1 \times 10^{-3}$  for  $\gamma N_2O_5$  around 0.03, and  $\pm 3.6 \times 10^{-5}$  3 for  $\gamma N_2 O_5$  around 0.01, with RH of 40%. The uncertainty increased with RH and would be 9% to 17% at  $\gamma N_2 O_5$  around 0.03 for RH ranging from 20% to 70% (Fig. 5b).

Sensitivity tests with the iterative model calculation were performed to evaluate the uncertainty associated with measurement accuracy of  $N_2O_5$  and VOCs, by varying the input  $N_2O_5$  concentrations and  $k_{NO3-VOC}$  in both modes. It is found that the  $N_2O_5$  measurement uncertainty of 25% (Tham et al., 2016; Wang et al., 2017) would translate into an uncertainty of 12% in the  $\gamma N_2O_5$  (shown in SI). The VOCs measurement uncertainty, however, has negligible influence on  $\gamma N_2O_5$  calculation. In previous flow tube method introduced by Bertram et al., (2009), they also explained that the homogeneous reaction was expected to be independent of the aerosol and non-aerosol modes and was thus can be canceled out in the calculation. Only strong atmospheric variation in VOC in short time period would influence the  $N_2O_5$  uptake measurement. The uncertainty introduced by the aerosol surface area measurement including aerosol loss influence would be propagated to an uncertainty in the  $\gamma N_2O_5$  calculation of 30%.

As mentioned in section 3, the use of mean residence time rather than RTD function by assuming an ideal reactor and ignoring diffusion and dispersion processes would also introduce uncertainties. In order to evaluate the magnitude of this bias, we have performed a simplified test by comparing a first-order loss rate from mean residence time with a residence time distribution range. Briefly, the mean concentration of  $N_2O_5$  at the exit the reactor could be expressed by:

$$\left[\overline{N_2 O_5}\right] = \int_0^\infty [N_2 O_5]_t E_t dt = \int_0^\infty [N_2 O_5]_0 e^{-kt} E_t dt \tag{9}$$

where  $[N_2O_5]t$  is the average concentration exit from the reactor between t and t + dt, E(t) is the residence time distribution function, and k is the first order loss rate coefficient of  $N_2O_5$ . The results showed that the first-order loss rate calculated from the distribution function was higher than that with a mean residence time, and was about 5% or 16% higher when the ratio of  $\frac{[N_2O_5]_t}{[N_2O_5]_0}$  was 0.6 or 0.2 in the flow tube system, respectively.

By incorporating all of these factors, the estimated total uncertainty is propagated to be 37% to 40% at  $\gamma N_2 O_5$  around 0.03 with 1000  $\mu m^2/cm^3$  Sa for RH ranging from 20% to 70%. "

Figure S1. Sensitivity test of the iterative model via varying input N2O5 and kNO3-VOC in both modes.

Page 9 Line 11: The retrieval of the N2O5 uptake coefficient is sensitive to a difference in NO3 reactivity between the aerosol on and off states. It would be helpful if the authors also stated how the difference in NO concentration between the on and off states impacted the retrieval.

**Response:** The NO titration effect would underestimate the uptake coefficient even when NO concentration is the same level between two modes, as shown in Fig. 7a. When NO concentration is higher, for example in aerosol ON mode, the measured  $N_2O_5$  concentration would be lower due to NO titration, thus overestimate the uptake coefficient if only compare exit concentration ratio of  $N_2O_5$  in two modes. In the ambient measurement case in Fig. 9a in section 7, we have compared the uptake coefficients derived from the iterative model method and exit-concentration ratio method when NO was fluctuated between aerosol on and off states. The determined  $\gamma N_2O_5$  was overestimated by 28% for the NO increase of about 1.5 ppbv. For comparison, we also chose different periods in aerosol existing state corresponding to different NO conditions in this case, and the iterative model derived similar loss rate constants and uptake coefficients, demonstrating the applicability of the iterative model in buffering against fluctuated NO.

The revised text is as follows,

"The overestimated  $\gamma N_2O_5$  from the exit-concentration ratio approach could be explained by the increased NO level (~ 1.5 ppbv) in the aerosol mode. For comparison, another two periods of data points in the March 21 case (Fig. 9a) with different NO levels were also selected to derive the  $k_{het}$ , and the results showed good consistency (0.0136-0.0140 s-1) (Fig S2 in SI), also demonstrating the applicability of the iterative model in buffering against fluctuated NO."

---

## Author Comment (AC1)

**Response to Anonymous Referee #1**

The authors present a flow tube measurement of the N2O5 uptake coefficient that is an extension of the work of Bertram, Riedel, and Thornton. The measurement system is described and it is similar to the earlier design. The main innovations presented here are a more detailed measurement of the residence time distribution in the flow tube and the application of an iterative box model to retrieve the uptake coefficient when ambient concentrations of NO, NO2 and O3 are high enough to make 2nd order reactions important in the flow tube. The authors also present ambient measurements of the uptake coefficient which are useful because these direct measurements are rare and limited geographically.

This is an important measurement and should be published in AMT with minor changes.

Suggestion: A method, complementary to the iterative box model analysis, would be to reduce the concentrations of the gas-phase interferers (NO, NO2, O3, VOCs) before the N2O5 addition using an actived-carbon scrubber that transmits aerosols, such as http://www.sunlab.com/denuders/.

**Response**: We thank the reviewer for his/her attention to this manuscript. We have made all of the suggested changes and/or made clarifications. The reviewer's comments are in black and our response is in blue and revised text in italic.

We also included the reviewer's suggestion of using active-carbon scrubber in the revised text, as follows,

"For future development, an activated-carbon scrubber in the inlet to reduce the gas-phase interferers (NO, NO2, O3, VOCs) but transmit aerosols could be a complementary approach to apply the flow tube system coupled with iterative box model analysis to even higher polluted conditions."

**Minor issues:**

1) Typically laboratory measurements of the uptake coefficient on synthetic aerosol are less than 0.04. Although some ambient analyses (Wagner et al. 2013, McDuffie et al. 2018) report uptake coefficients above 0.04 (upto 0.1) for a small subset of the data. It is not clear if these are artifacts of the analysis or real measurements of the uptake coefficient. Here the authors also report a direct measurements of uptake coefficients between 0.04 and 0.1. I would encourage the authors to address the discrepancy between laboratory measurement and their ambient measurements.

If they are real what is aerosol composition? Can the measured uptake be reproduced in the lab with synthetic aerosol?

**Response**: The discrepancy of uptake coefficient between laboratory measurement and ambient measurements via the indirect method have been reported by many researchers. This is also one of the motivations to improve the direct uptake coefficient measurement technique with an aerosol flow tube on ambient aerosols. We also conducted laboratory tests with (NH4)2SO4 aerosols by using the same system, and similar uptake coefficient around 0.02 was obtained under different

NO, NO2, and O3 conditions. The results are shown in the following table which has been added in the SI. The value is similar to previous laboratory results, which can serve as a validation of the applicability of the introduced system and also implies that the measured high uptake coefficient value is not due to the artificial of our aerosol flow tube system. The uptake coefficient on ambient aerosols in this study, however, was found to be more variable. During the campaign, the concentrations of water-soluble ions, organic/element carbon amount in the aerosol were also measured. However, it is hard to reproduce the complex aerosol composition as well as the mixing states in the laboratory. Thus, we will perform more studies and further analysis on the dependence of uptake coefficient on ambient aerosols compositions in the future works.

| No. | Initial NO 2
(ppb) | Initial O 3
(ppb) | Initial NO
(ppb) | Initial N 2 O 5
(ppb) | RH (%) | Sa ( $\mu m^2/cm^3$ ) | γ      |
|-----|----------------------------------|---------------------------------|---------------------|------------------------------------------------|--------|-----------------------|--------|
| 1   | 62                               | 57                              | 0                   | 2.1                                            | 25.1   | 848                   | 0.0226 |
| 2   | 62                               | 57                              | 5.0                 | 2.1                                            | 24.6   | 928                   | 0.0208 |
| 3   | 57                               | 106                             | 0                   | 4.3                                            | 22.9   | 965                   | 0.0182 |
| 4   | 57                               | 106                             | 5.0                 | 4.3                                            | 23.2   | 894                   | 0.0212 |
| 5   | 57                               | 106                             | 0                   | 4.3                                            | 48     | 1425                  | 0.0259 |

Table S1. Lab experiments with (NH4)2SO4 aerosols.

2) It is unclear what parameters were used in the uncertainty analysis. I suspect uncertainty due to the aerosol surface area measurement would be at least  $\pm/-25\%$ . In figure 9, there are not smooth exponential decay transitions between filter ON and OFF periods, so I suspect the uncertainty in the N2O5 measurement is significant.

On page 8 line 18, please list the key parameters and the uncertainty associated with them.

**Response**: Thanks for pointing this out. According to our and other previous studies, the uncertainty of the aerosol surface area measurement from the WPS system could be around 20-30% (Wang et al., 2017; Tham et al., 2018). The reason for not smooth exponential decay transitions between filter ON and OFF periods mainly due to air changes in ambient, flow turbulence when switching valves and diffusion/dispersion as a non-ideal reactor. As stated in our previous studies, the uncertainty of N2O5 measurement using the same instrument and same setup is 25% (Tham et al., 2016; Wang te al., 2017). We have revised this part to include these measurement uncertainties in the overall uncertainty estimation.

The revised text is as follows,

"The uncertainty of the aerosol surface area measurement was 20-30% (Wang et al., 2017; Tham et al., 2018)."

"In addition to  $k_{wall}$  being affected by RH, uncertainty in  $k_{aerosols}$  determination can also result from  $N_2O_5$  source variability,  $NO_3$  reactivity with VOCs, precision as well as accuracy associated with the measurement of all parameters. The long period of measurement cycle may also bring uncertainty due to concentrations variation in two operation modes. As described in Section 2.2,

the stability of the  $N_2O_5$  generation source was within  $\pm 2\%$  over an hour. In the present study, online VOCs were measured with a time resolution of one hour.  $A \pm 0.01$  s-1 variation of kNO3-VOC would lead to a single-point uncertainty in  $\gamma N_2 O_5$  of  $\pm 0.4 \times 10^{-3}$  for Sa = 1000  $\mu m^2/cm^3$ . NO reacts at a faster rate with NO3, having a larger impact on the  $\gamma N_2O_5$  calculation compared to VOCs. With a constrained real-time NO concentration, the iterative model can buffer against small NO changes. Stability of NO, NO2, O3, and N2O5 for a period of at least 5 minutes for each mode is required to ensure that the flow-tube reactor measurement and iterative model yield reasonable results. The measurement precision and variation of these species during each cycle might also introduce uncertainty in the iterative model calculation. The uncertainty in the  $\gamma N_2 O_5$ determination associated with  $k_{wall}$  changes, VOCs variation, and the variation of the different parameters during the measurement cycle was estimated with a Monte Carlo approach, as described in Groß et al. (2014), by assessing the uncertainty from individual key parameters (shown in Table 1) in the calculation model.  $\gamma N_2 O_5$  was found to be most sensitive to RH, which was closely related to  $k_{wall}$  as discussed before. Fig. 5a shows the partial uncertainty of  $\gamma N_2 O_5$ derived from Monte Carlo simulations with RH at 40%. The single-point uncertainty in  $\gamma N_2 O_5$  was estimated to be  $\pm 4.1 \times 10^{-3}$  for  $\gamma N_2O_5$  around 0.03, and  $\pm 3.6 \times 10^{-3}$  for  $\gamma N_2O_5$  around 0.01, with RH of 40%. The uncertainty increased with RH and would be 9% to 17% at  $\gamma N_2O_5$  around 0.03 for RH ranging from 20% to 70% (Fig. 5b).

Sensitivity tests with the iterative model calculation were performed to evaluate the uncertainty associated with measurement accuracy of  $N_2O_5$  and VOCs, by varying the input  $N_2O_5$  concentrations and  $k_{NO3-VOC}$  in both modes. It is found that the  $N_2O_5$  measurement uncertainty of 25% (Tham et al., 2016; Wang et al., 2017) would translate into an uncertainty of 12% in the  $\gamma N_2O_5$  (shown in SI). The VOCs measurement uncertainty, however, has negligible influence on  $\gamma N_2O_5$  calculation. In previous flow tube method introduced by Bertram et al., (2009), they also explained that the homogeneous reaction was expected to be independent of the aerosol and non-aerosol modes and was thus can be canceled out in the calculation. Only strong atmospheric variation in VOC in short time period would influence the  $N_2O_5$  uptake measurement. The uncertainty introduced by the aerosol surface area measurement including aerosol loss influence would be propagated to an uncertainty in the  $\gamma N_2O_5$  calculation of 30%.

As mentioned in section 3, the use of mean residence time rather than RTD function by assuming an ideal reactor and ignoring diffusion and dispersion processes would also introduce uncertainties. In order to evaluate the magnitude of this bias, we have performed a simplified test by comparing a first-order loss rate from mean residence time with a residence time distribution range. Briefly, the mean concentration of  $N_2O_5$  at the exit the reactor could be expressed by:

$$\left[\overline{N_2 O_5}\right] = \int_0^\infty [N_2 O_5]_t E_t dt = \int_0^\infty [N_2 O_5]_0 e^{-kt} E_t dt \tag{9}$$

where  $[N_2O_5]t$  is the average concentration exit from the reactor between t and t + dt, E(t) is the residence time distribution function, and k is the first order loss rate coefficient of  $N_2O_5$ . The results showed that the first-order loss rate calculated from the distribution function was higher

than that with a mean residence time, and was about 5% or 16% higher when the ratio of  $\frac{[N_2 O_5]_t}{[N_2 O_5]_0}$  was 0.6 or 0.2 in the flow tube system, respectively.

By incorporating all of these factors, the estimated total uncertainty is propagated to be 37% to 40% at  $\gamma N_2 O_5$  around 0.03 with 1000  $\mu m^2/cm^3$  Sa for RH ranging from 20% to 70%. "

Figure S1. Sensitivity test of the iterative model via varying input N2O5 and kN03-VOC in both modes.

3) Measurements of NO and VOCs are not described. Uncertainty due to reactions of NO3 with unmeasured VOCs should be bounded.

**Response**: The ambient NO was measured by another NOx analyzer while VOCs were measured by an online-GC. We have added this information in the manuscript:

"The ambient VOCs were determined using an online gas chromatograph (GC) coupled with a flame ionization detector (FID) and a mass spectrometer (MS). The VOCs concentrations were used to determine the  $k_{NO3-VOC}$  in the aerosol flow-tube system, which was treated as constant during the short-time period of flow tube measurement. The ambient NO level was measured by another chemiluminescence NOx analyzer (Thermo, Model 42i) equipped with a molybdenum converter."

The uncertainty due to reactions of NO3 with VOCs has been tested in a sensitivity test which used varied  $k_{NO3-VOC}$  as input. It shows that the uncertainty of  $k_{NO3-VOC}$  measurement could be negligible when comparing two modes. This information has been added in the text and SI, as details described in the above response.

4) The authors show that the residence time in the flow tube is a distribution (ranging over a factor of 2 in residence times), however in the iterative box model only the mean residence time is used. As the iterative box model likely depends in the residence time in a nonlinear way, the author should use a range of residence times in the iterative box model.

**Response**: Thanks for the valuable suggestion. Since it is very difficult to include the RTD function in the iterative model calculation, we have performed a simplified test to estimate the uncertainty that may arise from the use of mean residence time. The comparison results showed that the use of mean residence time might underestimate the loss rate coefficient by 5% to 16% for different conditions. We have added this information and more discussion in the revised text, as follows,

"The RTD function in Fig. 2 is clearly different from the ideal laminar flow reactor. Bertram et al. (2009) have suggested that the determined rate constant would be underestimated by up to 25% due to non-ideal plug flow condition. More discussion of the uncertainty in  $\gamma N_2O_5$  calculation associated with residence time distribution is presented in section 5."

"As mentioned in section 3, the use of mean residence time rather than RTD function by assuming an ideal reactor and ignoring diffusion and dispersion processes would also introduce uncertainties. In order to evaluate the magnitude of this bias, we have performed a simplified test by comparing a first-order loss rate from mean residence time with a residence time distribution range. Briefly, the mean concentration of  $N_2O_5$  at the exit the reactor could be expressed by:

$$\left[\overline{N_2 O_5}\right] = \int_0^\infty [N_2 O_5]_t E_t dt = \int_0^\infty [N_2 O_5]_0 e^{-kt} E_t dt \tag{9}$$

where  $[N_2O_5]t$  is the average concentration exit from the reactor between t and t + dt, E(t) is the residence time distribution function, and k is the first order loss rate coefficient of  $N_2O_5$ . The results showed that the first-order loss rate calculated from the distribution function was higher than that with a mean residence time, and was about 5% or 16% higher when the ratio of  $\frac{[N_2O_5]_t}{[N_2O_5]_0}$  was 0.6 or 0.2 in the flow tube system, respectively."

5) Have the authors measured particle losses in the flow tube? Diffusional and gravitational losses could be important. Could aerosol losses also be RH dependent? If so, please add a few sentences describing the results.

**Response**: Yes, we have measured the particle transmission in the introduced flow tube system using synthetic aerosols. This information has been added in the text, as follows,

"The transmission of aerosols in the flow tube was evaluated using laboratory-generated  $(NH_4)_2SO_4$  particles. The passing efficiency was around 50% for particles with a size of 20 nm, and more than 90% for particles larger than 100 nm. The total surface area loss in the flow tube was around 10-25%."

6) In figure 9, the periods chosen for analysis seems to be handpicked for stability. If different periods were chosen how would the results change?

**Response**: Since the mean residence time of the flow tube is more than 2 minutes, it is necessary to have at least 5 minutes of stable data for the calculation. We normally choose the 5 minutes data when all monitored parameters were relatively stable for each operation mode.

The fluctuation of the N2O5 signals in Fig.9a was mostly due to the variation of ambient air, such as change of NO levels. We have tried to use different time periods to perfume the calculation in the same case, as shown below, the different stable periods with different N2O5 and ambient NO level actually still gave similar results. We have included this information in the text and SI.

The revised text reads:

"For comparison, another two periods of data points in the March 21 case (Fig 9a) with different NO levels were also selected to derive the  $k_{het}$ , and the results showed good consistency (0.0136-0.0140 s-1) (Fig S2 in SI), also demonstrating the applicability of the iterative model in buffering against fluctuated NO."

Figure S2. sample case on Mar 21st, two stable data point under different NO level are chosen to calculate the N2O5 loss rate constant.

Technical issues:

Pg 3, line 26: How does the flow tube pressure relate to ambient pressure?

**Response:** Since the measurement in the present study was conducted at a low altitude site (60 m a.s.l), the ambient pressure was mostly close to 1 atm. We measured the pressure in the flow tube with a pressure meter occasionally, and the pressure did not show obvious change.

Pg 4, line 14: how much NO2 is added with the N2O5 addition?

**Response:** After dilution in sample air in the flow tube, the injection of NO2 concentration was 57 ppbv. This information has been added in the revised text, as follows,

"In typical experiment used in the present study, the input of the  $N_2O_5$  source to the top of flow tube contained 4.3 ppbv of  $N_2O_5$ , together with 106 ppbv of  $O_3$  and 57 ppbv of  $NO_2$ ."

Pg 7, line 4: This sentence is missing a subject

**Response:** Thanks for pointing out. The word "this" was added to the sentence. "this" here means consider NO3 and  $N_2O_5$  as one singular  $N_2O_5^*$  in the box model.

"Doing this also makes backward reaction simulation possible by avoiding unstable equilibrium in the box model."

Pg 7, line 20: Please give some more explanation about when non-physical results occur. When the uptake coefficient is small. When aerosol number is low? I expect that in a flow tube with high initial N2O5 the box model would work well in most cases.

**Response:** The low aerosol surface area and insignificant uptake could possibly result into negative uptake values when the heterogeneous loss on aerosols is small but the  $k_{NO3}$  or wall loss of N2O5 dominate the N2O5 loss in flow tube reactor and when the fluctuation of the wall loss due to temperature or RH is significant. Slightly higher initial N2O5 concentration could be useful to reduce the influence of these fluctuations but might also introduce other artifacts as suggested by Thornton (2003).

We have added more information in this part, as follows,

"This non-physical result might result from much larger fluctuations of  $k_{NO3}$  or  $k_{wall}$  in the system during each measurement cycle. When  $k_{aerosol}$  is small due to the low  $S_a$  or insignificant uptake, the  $k_{NO3}$  or  $k_{wall}$  may dominate the  $N_2O_5$  loss in flow tube reactor, and the fluctuations of  $k_{NO3}$  or  $k_{wall}$  due to the air mass or temperature/RH changes would bias the  $k_{aerosol}$  determination and led to large uncertainty or negative values. This situation often occurred under conditions of fresh NO emission; more discussion of the influence of NO is presented in section 6."

Pg 8, line 5: please add 'respectively'

**Response:** The sentence has been revised as,

"This result would translate into an uncertainty of  $(\pm 0.15 \times 10^{-3})$  to  $(\pm 2.4 \times 10^{-3})$  in  $\gamma N_2O_5$  with RH of 20% to 70%, respectively, and a Sa of 1000  $\mu m^2/cm^3$ ."

Pg 9 line 9: typo 'ere'

Response: Corrected.

"The  $N_2O_5$  regeneration effect on  $\gamma N_2O_5$  calculation was significant when  $O_3$  and  $NO_x$  levels in the ambient air are high."

Pg 9, line 16: Could you summarize the potential artifacts.

**Response:** Thornton (2003) reported a higher uptake coefficient obtained when using initial  $N_2O_5$  of 6 ppbv than with 30 ppbv. They suggested that the artifacts could be the particulate  $NO_3^-$  formed via  $N_2O_5$  hydrolysis inhibiting further ionization of  $N_2O_5$  when initial  $N_2O_5$  is too high.

Pg. 10 line 27: missing 'the', 'in aerosol mode'

Response: Corrected.

"The overestimated  $\gamma N_2 O_5$  from the exit-concentration ratio approach could be explained by the increased NO level (~ 1.5 ppbv) in the aerosol mode."